# Understanding the Pathways from Prenatal and Post-Birth PM_2.5_ Exposure to Infant Death: An Observational Analysis Using US Vital Records (2011–2013)

**DOI:** 10.3390/ijerph19010258

**Published:** 2021-12-27

**Authors:** Aayush Khadka, David Canning

**Affiliations:** 1Department of Family and Community Medicine, University of California San Francisco, San Francisco, CA 94110, USA; 2Department of Global Health and Population, Harvard T. H. Chan School of Public Health, Boston, MA 02115, USA; dcanning@hsph.harvard.edu

**Keywords:** early life exposures, fine particulate matter, PM_2.5_ exposure, prenatal air pollution, post-birth air pollution, infant mortality, Structural Equation Model

## Abstract

We studied the relationship of prenatal and post-birth exposure to particulate matter < 2.5 μm in diameter (PM_2.5_) with infant mortality for all births between 2011 and 2013 in the conterminous United States. Prenatal exposure was defined separately for each trimester, post-birth exposure was defined in the 12 months following the prenatal period, and infant mortality was defined as death in the first year of life. For the analysis, we merged over 10 million cohort-linked live birth–infant death records with daily, county-level PM_2.5_ concentration data and then fit a Structural Equation Model controlling for several individual- and county-level confounders. We estimated direct paths from the two exposures to infant death as well as indirect paths from the prenatal exposure to the outcome through preterm birth and low birth weight. Prenatal PM_2.5_ exposure was positively associated with infant death across all trimesters, although the relationship was strongest in the third trimester. The direct pathway from the prenatal exposure to the outcome accounted for most of this association. Estimates for the post-birth PM_2.5_–infant death relationship were less precise. The results from our study add to a growing literature that provides evidence in favor of the potential harmful effects on human health of low levels of air pollution.

## 1. Introduction

Although several studies have investigated the relationship between early life ambient air pollution exposure and infant mortality, the evidence about the importance of prenatal versus post-birth exposure is mixed [1,2]. Some studies have found that exposure to higher levels of post-birth air pollution increases the risk of infant death but increased prenatal air pollution exposure does not [3,4]. In contrast, other studies have found that higher levels of prenatal air pollution exposure increases the risk of infant death, but post-birth air pollution exposure does not [5,6]. Furthermore, some studies have also suggested that both prenatal and post-birth air pollution exposure may increase the risk of infant mortality [7,8].

There are plausible biological mechanisms to link both prenatal and post-birth air pollution exposure to infant death. For instance, there is very strong evidence linking prenatal air pollution exposure with increased risk of adverse birth outcomes such as preterm birth and low birth weight [9]. Such a relationship would suggest that prenatal air pollution may be linked to infant death because prematurity and low birth weight are two of the most important drivers of infant mortality in the US [10]. At the same time, there are biologically plausible direct pathways from prenatal air pollution exposure to infant mortality as well: for example, exposure to fine particulate matter can cause imbalances in an individual’s autonomic nervous system and result in oxidative stress, both of which when experienced by pregnant people can affect maternal and fetal health, which in turn can have implications for infant health [11,12]. Fine particulate matter can also traverse the placental barrier, thus directly affecting the health of the developing fetus, which in turn could lead to worse infant health outcomes and, subsequently, infant death [13]. In terms of post-birth exposure, infants who are exposed to high levels of air pollution may demonstrate similar pathophysiological responses as adults which, in combination with their more immature immune and lung systems, could increase their risk of morbidity and death as well [14,15].

Given the biological plausibility, the mixed nature of the evidence on the relationship of prenatal and post-birth air pollution exposure with infant death may be due to several reasons. First, studies are often conducted in different geographic settings and over different time periods, which have important implications for the type and level of pollution exposure. Second, the exposure may be measured differently across different studies, with some using modeled estimates, while others use distance-weighted estimates of pollution directly from monitors. Finally, analytic strategies may be different across different studies, which could potentially affect the results. For example, variables on the causal pathway, such as gestational length and birth weight, may be adjusted for without accounting for potential collider stratification bias [1,9,16,17,18].

In this study, we revisit the relationship between early life ambient air pollution exposure and infant mortality in the US with regard to particulate matter less than 2.5 μm in diameter (PM_2.5_). Specifically, we aim to estimate direct pathways from prenatal and post-birth PM_2.5_ to infant death as well as indirect pathways from the prenatal exposure to the outcome through preterm birth and low birth weight. Investigating the pathways in such a manner not only allows us to compare the relative associations of prenatal versus post-birth exposure but also helps us understand the relative importance of the indirect versus direct pathway in terms of the prenatal PM_2.5_–infant death relationship. PM_2.5_ is an important pollutant to study in this context because of its known association with various indicators of fetal health such as fetal growth and organ development as well as its relationship with infant morbidity, especially respiratory diseases [19,20,21].

## 2. Materials and Methods

### 2.1. Data Sources

To define our exposure variables, we used modeled estimates of daily, population-weighted mean PM_2.5_ concentration at the county level within the conterminous US between 2009 and 2014. These publicly available estimates are provided by the Centers for Disease Control and Prevention and are based on the Environmental Protection Agency’s (EPA) Downscaler model [22,23]. The Downscalar model fuses together modeled estimates of air pollution concentration from the EPA’s Community Multi-Scale Air Quality model with data drawn directly from air pollution monitors [23,24].

To define our outcome and mediators, we used restricted access, cohort-linked birth–infant death data between 2011 and 2013. These data were provided by the National Center for Health Statistics and represent the universe of live births and infant deaths for children born between 2011 and 2013. These data also contain information on the pregnant person’s county of residence at delivery as well as characteristics related to parental demographics, pregnancy, delivery, and infant death.

We collected information on county-level confounders from several publicly available data sources. We extracted information on monthly average temperature and rainfall from the National Oceanic and Atmospheric Administration [25]. From the US Census Bureau, we extracted information on annual county-level demographic and economic variables including racial composition, total population, poverty rate, and median household income [26,27]. Similarly, from the Bureau of Labor Statistics, we extracted information on monthly unstandardized unemployment rate at the county level [28]. Finally, from the Centers for Medicare & Medicaid Services, we extracted information on the annual number of physicians in a county [29].

### 2.2. Outcome Definition

Our primary outcome was infant death, which we defined as an indicator variable that takes the value one if a baby dies in the first year of life for any reason and zero otherwise.

### 2.3. Exposure Definition

We defined prenatal air pollution exposure as the average PM_2.5_ concentration in each trimester in the pregnant person’s county of residence at delivery. Disaggregating the prenatal exposure by trimester reflected results from the literature which suggest that the prenatal air pollution–adverse birth outcome relationship varies by pregnancy trimester [30]. The first trimester was defined as the first three months following the date of conception. The second and third trimesters were defined as the three months following the first and second trimesters, respectively. As such, the prenatal exposure period spanned a total of nine months from the date of conception. We defined the prenatal exposure over a nine-month period as opposed to the actual length of gestation because the latter is a function of prenatal air pollution [9].

We similarly defined post-birth exposure as the average PM_2.5_ concentration in the pregnant person’s county of residence in the 12-month period following the end of the nine-month prenatal period. We defined post-birth exposure over a 12-month period since the number of days alive in the first year of life may be a function of air pollution exposure as well. We also defined the post-birth exposure from the end of the nine-month prenatal period as opposed to the date of birth to ensure that the prenatal and post-birth periods did not overlap temporally.

### 2.4. Mediator Definition

We used preterm birth and low birth weight as the two mediators in our analysis. Preterm birth was coded as an indicator variable that equaled one if the obstetric/clinical estimate of gestational length was less than 37 weeks and zero otherwise. Low birth weight was coded as an indicator variable that equaled one if birth weight was reported to be less than 2500 g and zero otherwise.

### 2.5. Constructing the Analytic Dataset

To construct our analytic sample, we applied the following inclusion criteria to the cohort-linked birth–infant death records: (1) birth records were collected using the latest revision of the birth certificate (i.e., the 2003 birth certificate revision, which had revised a number of instruments for collecting data, including maternal and paternal race as well as cigarette smoking during pregnancy); and (2) pregnant person’s county of residence had daily PM_2.5_ concentration information available.

To merge the birth–infant death records with the exposure data, we began by estimating the date of conception using information on the year, month, day of week of birth, and the obstetric/clinical estimate of gestational length. Specifically, using the day of the week, month, and year of birth information, we randomly assigned each birth to a date of birth by assuming a uniform probability distribution over the day of week within any given month–year. Then, we subtracted the length of gestation from the estimated date of birth to get our estimate of the date of last menstrual period (LMP). Finally, we assigned the date of conception for each live birth observation by adding two weeks to the LMP date under the assumption that conception occurs, on average, two weeks after the LMP.

Then, we merged the birth–infant death records with the PM_2.5_ data based on each pregnant person’s reported county of residence and the date of conception. Similarly, we merged the monthly temperature, rainfall, and unemployment data with the live birth–infant death records based on the county of residence and month and year of conception. Finally, we merged data on annual population, poverty rate, housing value, and healthcare access based on the county of residence and year of conception.

### 2.6. Structural Equation Model

We applied a Structural Equation Modeling (SEM) framework to estimate the direct and indirect pathways from prenatal air pollution exposure to infant death as well as the direct path from post-birth air pollution exposure to the outcome. We also modeled a direct path from prematurity to low birth weight, since birth weight is a function of gestational length [31]. An SEM framework is appropriate for this analysis because it allows us to estimate all pathways simultaneously, which improves statistical power and allows us to estimate total direct and indirect associations easily [32]. A graphical representation of our SEM is presented in Figure 1.

To identify the direct path from prenatal PM_2.5_ exposure to adverse birth outcomes, we controlled for a variety of individual, pregnancy, and county-level covariates. At the individual level, we controlled for the pregnant person’s age, race, highest level of education, marital status, parity, pre-pregnancy smoking behavior, and average PM_2.5_ exposure in the nine months prior to conception. We also controlled for father’s age, race, and highest level of education. In terms of pregnancy characteristics, we controlled for the method of payment for delivery, multiple birth, and child’s sex. Finally, at the county level, we controlled for average temperature, precipitation, and unemployment rate over the prenatal period. We also controlled for annual county racial composition, poverty rate, median housing value, and number of physicians per 1000 individuals. To identify the direct path from post-birth PM_2.5_ exposure to infant mortality, we controlled for these same variables.

To identify the direct path from adverse birth outcomes to infant mortality as well as the path from preterm birth to low birth weight, we additionally controlled for cigarette smoking during pregnancy. We did not control for variables such as number of prenatal care visits because they may plausibly fall on the direct pathway from prenatal air pollution exposure to infant death. As such, controlling for these variables would lead to potentially biased estimates of the direct association between prenatal PM_2.5_ exposure and infant death.

Finally, to account for unobserved time-invariant county-specific sources of confounding, we allowed for county fixed effects in all models by applying the fixed effects transformation (i.e., demeaning the data at the county level) [33]. In addition, we flexibly modeled time trends in the outcome in all models by including month-of-year fixed effects and a linear, county-specific time trend over the study period. Table A1 presents detailed definitions of all covariates used in the SEM.

### 2.7. Statistical Analysis

Before fitting the SEM, we estimated summary statistics of the exposures, mediators, covariates, and outcome. For ease of illustrating summary statistics, we used average PM_2.5_ concentration over the entire nine-month prenatal period as opposed to each trimester.

To prepare our data for the SEM, we imputed missing values in the analytic sample five times by assuming that the data were missing at random and that the observed and unobserved data followed a multivariate normal distribution. We estimated the SEM in each imputed dataset using the standard maximum likelihood method with an identity link function in all equations. Using the identity link function allowed us to easily estimate direct, indirect, and total associations. We corrected our standard errors by using Huber’s cluster-robust variance estimator at the county-level, which allowed us to account for correlated outcomes within a county [34]. We combined the estimates from the SEM across all imputed datasets by using Rubin’s rules [35].

The literature on air pollution and infant mortality suggests that the exposure–outcome relationship may be non-linear [36]. To account for such potential non-linearities, we categorized both the prenatal and post-birth exposure as <8 μg/m^3^; 8–10 μg/m^3^; 10–12 μg/m^3^; and, ≥12 μg/m^3^. These cutoffs reflected the distribution of the exposure variables in our dataset (Appendix A
Figure A1) and captured the annual air quality guidance of the World Health Organization (WHO; threshold = 10 μg/m^3^) and the EPA (threshold = 12 μg/m^3^) [37,38].

### 2.8. Robustness Checks

We conducted two robustness checks. For both robustness checks, we redefined the post-birth exposure variable and re-fit the SEM. Specifically, we redefined the post-birth exposure as average PM_2.5_ concentration in the one month and two months following the end of the nine-month prenatal period. This reflected the fact that over three-quarters of all infant deaths in the US occurred within the first two months of life (Table A2).

### 2.9. Software

We used Stata/IC 15 to clean the data, create the analytic sample, and estimate the SEM model [39]. We used RStudio to estimate values of the exposure in the preconception, prenatal, and post-birth periods, create all the figures, and impute the data (package: Amelia II) [40,41].

### 2.10. Ethical Statement

This study was exempted from human subjects review by the Institutional Review Board at the Harvard T. H. Chan School of Public Health as per regulations found at 45 CFR 46.104(d) (4).

## 3. Results

Our analytic sample consisted of 10,017,357 live births and 58,913 infant deaths in 3053 counties across 48 states in the conterminous US. In addition to Alaska and Hawaii, Washington DC was also excluded from our analysis due to a lack of air pollution data. Appendix A Figure A2 shows that the primary reason for excluding live birth and infant death observations from our analytic sample was births not being recorded using the 2003 revision of the birth certificate. Missing observations were relatively rare (<10%) except in the case of the father’s age, race, and education, where missingness was between 12% and 15% (Table A3).

Figure 2 shows the annual average PM_2.5_ concentration in counties where a conception occurred between 2010 and 2013, which were the earliest and latest conception years in our analytic sample. The average annual PM_2.5_ concentration decreased over the study period from 9.20 μg/m^3^ in 2010 to 8.34 μg/m^3^ in 2013. PM_2.5_ concentration also varied substantially by region: counties in the Interior Midwest, the South, and the Southwest experienced the highest levels of air pollution, while counties in the Great Plains experienced the lowest levels of air pollution.

In Figure 3, panels (a) and (b) respectively show unadjusted averages of infant mortality by the four categories of prenatal and post-birth PM_2.5_ exposure. Prenatal exposure here is defined as the average PM_2.5_ concentration over the entire nine-month prenatal period as opposed to the average concentration over each trimester. Infant mortality was increasing with prenatal air pollution over the study period; however, in terms of the post-birth exposure, infant mortality increased over the first three air pollution categories and decreased in the final category. Panels (c,d), which present unadjusted proportions of preterm birth and low birth weight by categories of the prenatal exposure variable, follow a similar pattern with the proportion of these two adverse birth outcomes increasing over the first three air pollution categories and then decreasing.

Approximately three-quarters of the pregnant people in our sample were between 20 and 34 years, a majority were non-Hispanic white, and more than 80% had a high school degree or higher (Table 1). In addition, approximately 48% and 43% of mothers paid for their deliveries using private insurance and Medicaid, respectively. When disaggregating these covariates by categories of the prenatal and post-birth exposure, we find that relative to the lowest air pollution category, the highest category had a lower proportion of individuals with a high school degree or more, higher proportion of deliveries paid for using Medicaid, and a higher average poverty rate (Table A4 and Table A5). Additionally, in terms of the prenatal exposure, the highest air pollution category had a higher proportion of Non-Hispanic Blacks and Hispanics relative to the lowest category (Table A4).

Table 2 presents the primary results from our analysis. The average Standardized Root Mean Square Residual (SRMR) of the overall model across all five imputed datasets was less than 0.01, which suggests good fit with the data. Panel D presents associations of trimester-wise prenatal PM_2.5_ exposure and post-birth PM_2.5_ exposure with infant death. There are two key take-aways from the results presented in this panel: first, the association between prenatal PM_2.5_ exposure and infant death is positive in all three trimesters but appears to be strongest in the third trimester. For instance, in the third trimester, relative to the reference exposure category of less than 8 μg/m^3^, being exposed to, on average, 8–10 μg/m^3^, 10–12 μg/m^3^, and ≥12 μg/m^3^ of PM_2.5_ concentration was associated with a 0.03 percentage point (95 percent confidence interval: 0.02, 0.05), 0.07 percentage point (0.05, 0.09), and 0.1 percentage point (0.06, 0.13) increase in the risk of infant death. These point estimates are generally larger in magnitude than those estimated for the first and second trimesters, although there is substantial overlap in the 95% confidence intervals. The second take-away is that the risk differences associated with post-birth PM_2.5_ exposure are less precisely estimated: for instance, relative to the reference category of less than 8 μg/m^3^, risk difference estimates related to being exposed to 10–12 μg/m^3^ of PM_2.5_ concentration over the study period were compatible with anything between a 0.03 percentage point decrease and a 0.05 percentage point increase in the risk of infant death.

Panels A and B of Table 2 show the association of prenatal PM_2.5_ exposure by trimester with preterm birth and low birth weight, respectively. The results presented in these panels suggest that increased PM_2.5_ exposure in the second and third trimesters was especially strongly associated with increased risk of experiencing the two adverse birth outcomes. For example, the second trimester results in Panel A suggest that being exposed to, on average, 8–10 μg/m^3^, 10–12 μg/m^3^, and ≥12 μg/m^3^ of PM_2.5_ concentration was associated with a 0.37 percentage point (0.25, 0.49), 0.62 percentage point (0.48, 0.77), and 0.74 percentage point (0.56, 0.92) increase in the risk of prematurity relative to the reference exposure category of less than 8 μg/m^3^. Panel C presents the association between preterm birth and low birth weight, while Panel E presents the association of the two adverse birth outcomes with infant death. Panel C suggests that preterm birth and low birth weight are strongly, positively associated; similarly, Panel E shows that both adverse birth outcomes are strongly associated with increased risk of infant mortality.

Table 3 shows the direct and indirect associations of prenatal PM_2.5_ exposure with infant death (Panel A) as well as the direct association of post-birth PM_2.5_ exposure (Panel B) with our primary outcome. The total association between prenatal air pollution and infant death is positive across all three trimesters; however, as shown in Table 2, the magnitude of the exposure–outcome relationship appears to be strongest in the third trimester. For instance, relative to the lowest exposure category, being exposed to ≥12 μg/m^3^ of PM_2.5_ concentration in the third trimester was associated with a 0.15 percentage point (0.11, 0.18) increase in the risk of infant death; in the first trimester, the estimated risk difference was compatible with any value between −0.01 percentage points and 0.07 percentage points. The proportion mediated across all three trimesters is less than 50%, which suggests that the majority of the association between prenatal PM_2.5_ and infant death is being driven through the direct pathway between the exposure and the outcome.

Table A6 and Table A7 present results from our robustness checks in which we redefined the post-birth exposure over a one- and two-month period, respectively. In contrast to our primary results, when we defined the post-birth exposure over a one-month period following the end of the nine-month prenatal period, we were able to estimate a positive association more precisely between post-birth air pollution and infant death across all levels of the exposure. However, as in our primary results, the post-birth air pollution–infant death relationship was less precisely estimated when we defined the post-birth exposure over a two-month period. Table A8 and Table A9 agree with our primary results in that they suggest that a majority of the association between the prenatal air pollution exposure and infant death is driven by the direct path from the exposure to the outcome and not the path through the two adverse birth outcome mediators.

## 4. Discussion

We studied the associations of prenatal and post-birth exposure to PM_2.5_ with the risk of infant death for all births that took place in the US between 2011 and 2013. We found that increased prenatal exposure to PM_2.5_ across all three trimesters was associated with an increased risk of infant death, although associations were strongest for air pollution exposure in the third trimester. We also found that the majority of the association between prenatal PM_2.5_ and infant death was driven by the direct path from the exposure to the outcome rather than the indirect paths through preterm birth and low birth weight. We were able to estimate the association less precisely between post-birth PM_2.5_ and infant death in our primary analysis; however, results from our robustness checks suggest the possibility of a positive association between the post-birth air pollution exposure and the infant death outcome.

Overall, the results from our study provide evidence in favor of increased air pollution exposure during gestation and possibly during the first year of life being harmful in terms of infant health, even at levels below the threshold set by the WHO and the EPA. This implication is consistent with recent findings from Di et al. (2018), which suggested that increased PM_2.5_ exposure below the EPA standard was associated with an increased risk of death among older Americans [42]. It is also nominally consistent with the results from Chay and Greenstone (2003), who demonstrated a positive association between total suspended particulates and infant death at levels below the EPA mandated threshold, although their analysis used data from the early 1980s when the EPA threshold was different from what it is currently [8].

Additionally, our result that over 50% of the association between prenatal PM_2.5_ exposure and infant death can be explained by the direct pathway suggests that there are mechanisms other than preterm birth and low birth weight by which in utero PM_2.5_ exposure may affect the risk of infant death. Previous studies have suggested that prenatal air pollution exposure may be linked with intrauterine growth retardation or congenital heart defects, all of which could affect the risk of infant mortality [43,44]. Future research should consider explicitly characterizing these pathways.

Finally, the discrepancy between the results from our primary analysis and robustness checks for the post-birth exposure highlights the complexities in defining the exposure in studies investigating health outcomes in early life. A strength of our analysis relative to the literature is that we define our exposure variables independent of the actual length of gestation or time alive. While this allows us to avoid defining the exposure using metrics that themselves may be a function of the exposure, it also introduces error into the exposure variables, which could affect both the point estimate and its associated standard error. Future research should consider determining the most appropriate methods of defining in utero and post-birth exposures.

Our study improves on the existing literature in several ways: first, we used high quality air pollution and infant death data with wide geographical coverage. Second, following lessons from the causal mediation literature, we carefully controlled for exposure–outcome, exposure–mediator, and mediator–outcome confounders [45]. We included county fixed effects in all our models, which allowed us to control for any time-invariant sources of confounding at the county level. We also modeled the time trend flexibly at the month-of-year level and the county level, which allowed us to net out any trends in the outcome. Third, we used an SEM framework to measure the different pathways from prenatal and post-birth PM_2.5_ exposure to infant death. To the best of our knowledge, our paper is one of the first to use SEM to understand this system of relationships.

Nevertheless, our analysis is still subject to several limitations. First, we were unable to disaggregate the pollution measure by type of PM_2.5_ pollutant. Previous studies have shown that there is a high degree of heterogeneity in terms of particulate matter composition in the US by region and season [46]. Furthermore, studies have also suggested that early life exposure to carbonaceous PM_2.5_ changes the risk of infant death, but sea salt or mineral dust does not [47]. Our use of county fixed effects somewhat addresses this issue by ensuring that we only use within-county variation in the exposure for our analysis.

Second, we did not include other pollutants in our model due to a lack of high-quality, daily data on them. Pollutants such as ozone, carbon monoxide, and nitrogen dioxide have been shown to be associated with infant death, while other studies have indicated that these pollutants may be correlated with PM_2.5_ levels as well [48,49,50,51,52,53,54,55,56]. Our inability to control for these pollutants means that we cannot rule out that the associations we present reflect the relationship between general pollution levels and infant death.

Third, the fact that air pollution data were only available at the county level may have introduced measurement error into our exposure variable. However, this issue is somewhat mitigated by the fact that the air pollution exposure is weighted based on the county’s population.

Fourth, in defining the exposure variables, we only have the pregnant individual’s county of residence at the time of delivery. Therefore, we are not able to account for changes in county of residence during pregnancy or in the first year of the child’s life. In addition, we are also not able to account for movement across counties for work or visits that might affect an individual’s exposure levels. Although there are relatively few studies examining the movement of pregnant individual’s during pregnancy, the existing literature suggests that when pregnant people do move, the median distance travelled is under 10 km [57,58]. This suggests that any issues due to the movement of individuals during pregnancy may be quite small.

Fifth, we were unable to account for indoor air pollution in our analysis. Despite being implicated in increasing the risk of several morbidities such as respiratory ailments and heart disease, we have relatively low understanding of the level of indoor air pollution in the US [59,60]. Future studies should consider studying the interaction between the level and type of indoor and outdoor air pollutants on infant morbidity and mortality.

Finally, a small but growing literature suggests that increased in utero air pollution exposure may increase the risk of pregnancy loss [61,62]. Since we use vital statistics data, our study effectively conditions on live births to analyze the relationship between prenatal air pollution exposure and infant death. As such, our estimates may be subject to a form of selection bias known as live birth bias [63,64,65]. Future studies should further investigate the degree to which differential fetal loss by levels of prenatal air pollution exposure impacts investigations of the relationship between early life air pollution exposure and infant health outcomes.

## 5. Conclusions

We showed that prenatal PM_2.5_ exposure was positively associated with infant mortality, with much of the association being driven by the direct path from the exposure to the outcome. Estimates for the relationship between post-birth PM_2.5_ exposure and infant mortality were positive but less precisely estimated. Our study motivates the need to understand if specific PM_2.5_ pollutants are responsible for these observed relationships or whether the associations we present represent general air pollution effects. Importantly, our study also adds to the literature that provides evidence in favor of low levels of air pollution being potentially harmful for human health. As more high-quality evidence on this topic accumulates, it may motivate the need to rethink the existing air pollution standards set by the WHO and the EPA.

## Figures and Tables

**Figure 1 ijerph-19-00258-f001:**
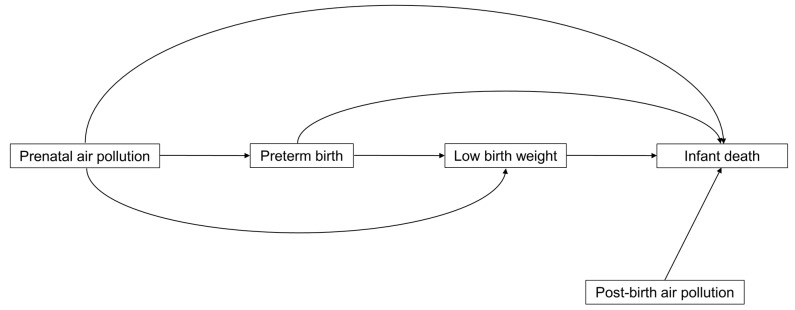
Graphical representation of the Structural Equation Model.

**Figure 2 ijerph-19-00258-f002:**
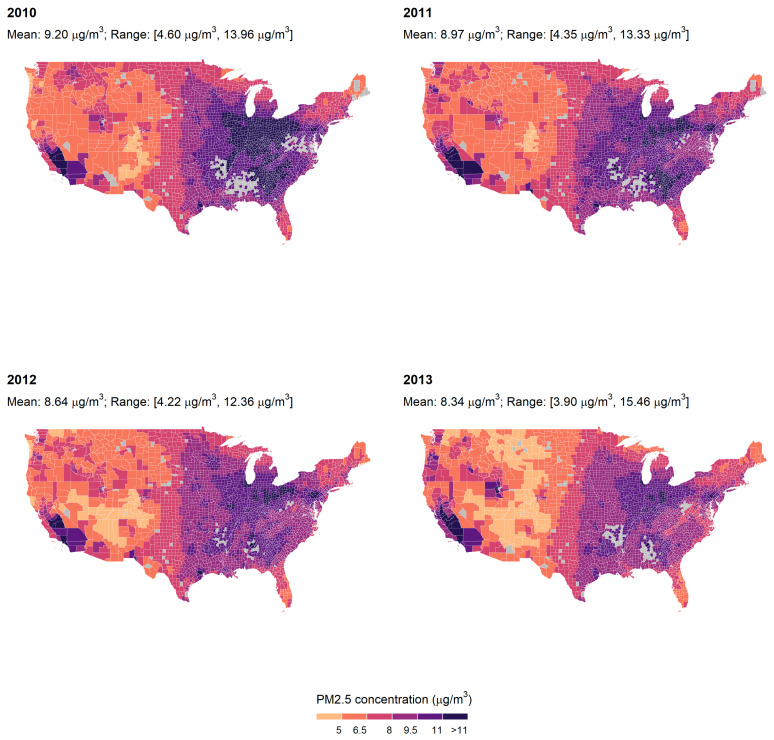
Average annual PM_2.5_ concentration by conception year in counties in which at least one conception occurred. Note: Each panel represents average PM_2.5_ concentration in counties in the conterminous US in the indicated year. Gray-colored counties are counties where no birth occurred in the respective year.

**Figure 3 ijerph-19-00258-f003:**
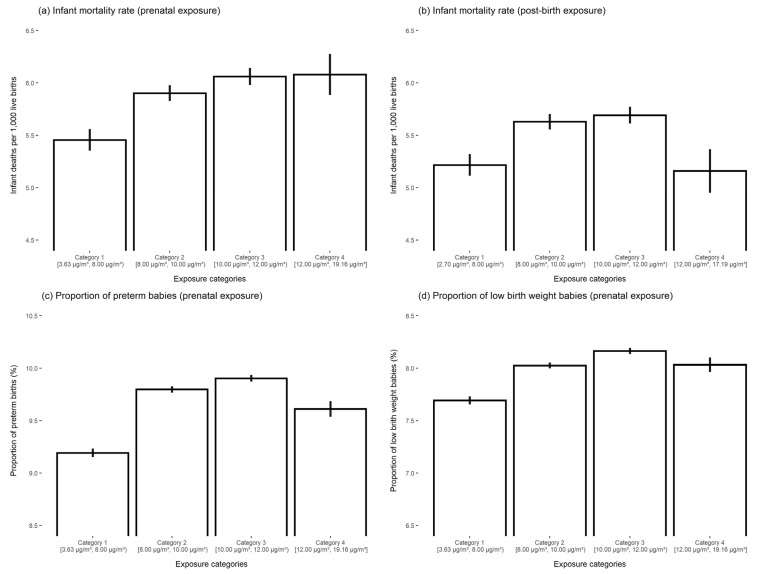
Unadjusted averages of infant mortality, preterm birth, and low birth weight by categories of the prenatal and post-birth PM_2.5_ exposure. Notes: Prenatal exposure is defined as average PM_2.5_ concentration in the pregnant person’s county of residence over the entire nine-month prenatal period. We define prenatal exposure over the entire nine-month period for ease of presentation. The categories of both prenatal and post-birth exposure are <8 μg/m^3^; 8–10 μg/m^3^; 10–12 μg/m^3^; and, ≥12 μg/m^3^. Panel (**a**) shows unadjusted infant deaths per 1000 live births by categories of the prenatal exposure variable. Panel (**b**) shows unadjusted infant deaths per 1000 live births by categories of the post-birth exposure variable. Panels (**c**,**d**) show unadjusted averages of preterm birth and low birth weight by categories of the prenatal exposure variable.

**Table 1 ijerph-19-00258-t001:** Distribution of individual, delivery, and county-level covariates in the analytic sample.

	Mean (SD)
	Overall
Individual-level variables	
Preconception PM_2.5_ concentration (μg/m^3^)	9.67 (1.76)
Mother’s age	
≤19 years	7.83 (26.86)
20–24 years	23.23 (42.23)
25–29 years	28.65 (45.21)
30–34 years	25.59 (43.64)
35–39 years	11.81 (32.27)
40–44 years	2.7 (16.21)
≥45 years	0.19 (4.36)
Mother’s race	
Non-Hispanic white	55.33 (49.72)
Non-Hispanic black	14.42 (35.13)
Non-Hispanic other	6.75 (25.09)
Hispanic	23.5 (42.4)
Mother’s education	
No high school	16.8 (37.38)
High school/some college	46.47 (49.87)
College or more	36.74 (48.21)
Mother is married	59.54 (49.08)
Mother smoked cigarettes pre-pregnancy	11.71 (32.16)
Parity	
First child	32.97 (47.01)
Second child	28.44 (45.11)
Third or more child	38.59 (48.68)
Payment source for delivery	
Medicaid	43.16 (49.53)
Private insurance	47.59 (49.94)
Self-pay	4.25 (20.16)
Other	5 (21.8)
Child born female	48.82 (49.99)
Singleton delivery	96.56 (18.22)
Father’s age	
≤19 years	3.05 (17.2)
20–24 years	15.3 (36)
25–29 years	25.55 (43.61)
30–34 years	28.41 (45.1)
35–39 years	17 (37.56)
40–44 years	7.29 (26)
≥45 years	3.41 (18.14)
Father’s race	
Non-Hispanic white	56.08 (49.63)
Non-Hispanic black	12.75 (33.36)
Non-Hispanic other	7.44 (26.24)
Hispanic	23.73 (42.54)
Father’s education	
No high school	15.96 (36.62)
High school/some college	48.62 (49.98)
College or more	35.43 (47.83)
County-level variables	
Average temperature during pregnancy	58.05 (9.54)
Average precipitation during pregnancy	3.06 (1.54)
Average unemployment during pregnancy	8.51 (2.48)
County racial composition	
Non-Hispanic white	61.99 (22.11)
Non-Hispanic black	12.3 (12.62)
Non-Hispanic other	5.6 (6.14)
Hispanic	18.26 (18.22)
Average poverty rate	16.1 (5.47)
Median household income (USD)	52,311.3 (13,252.36)
Physicians per 1000 individuals	0.35 (0.85)

Notes: SD = standard deviation. USD = United States dollars.

**Table 2 ijerph-19-00258-t002:** Estimates of the association between prenatal PM_2.5_ exposure, post-birth PM_2.5_ exposure, preterm birth, low birth weight, and infant mortality from the Structural Equation Model.

	Percentage Point Change	95% Confidence Interval
Panel A: Direct association between prenatal PM_2.5_ exposure and preterm birth		
First trimester		
Ref: <8 μg/m^3^		
[8.00–10.00 μg/m^3^)	0.15	[0.03, 0.26]
[10.00–12.00 μg/m^3^)	0.08	[−0.11, 0.27]
[12.00–19.16 μg/m^3^]	−0.08	[−0.33, 0.18]
Second trimester		
Ref: <8 μg/m^3^		
[8.00–10.00 μg/m^3^)	0.37	[0.25, 0.49]
[10.00–12.00 μg/m^3^)	0.62	[0.48, 0.77]
[12.00–19.16 μg/m^3^]	0.74	[0.56, 0.92]
Third trimester		
Ref: <8 μg/m^3^		
[8.00–10.00 μg/m^3^)	0.38	[0.26, 0.49]
[10.00–12.00 μg/m^3^)	0.71	[0.55, 0.87]
[12.00–19.16 μg/m^3^]	1.02	[0.82, 1.21]
Panel B: Direct association between prenatal PM_2.5_ exposure and low birth weight		
First trimester		
Ref: <8 μg/m^3^		
[8.00–10.00 μg/m^3^)	0.07	[0.01, 0.12]
[10.00–12.00 μg/m^3^)	0.08	[0, 0.15]
[12.00–19.16 μg/m^3^]	0	[−0.09, 0.08]
Second trimester		
Ref: <8 μg/m^3^		
[8.00–10.00 μg/m^3^)	0.11	[0.06, 0.17]
[10.00–12.00 μg/m^3^)	0.15	[0.08, 0.23]
[12.00–19.16 μg/m^3^]	0.23	[0.14, 0.31]
Third trimester		
Ref: <8 μg/m^3^		
[8.00–10.00 μg/m^3^)	0.13	[0.07, 0.18]
[10.00–12.00 μg/m^3^)	0.22	[0.15, 0.29]
[12.00–19.16 μg/m^3^]	0.35	[0.26, 0.43]
Panel C: Direct association between preterm birth and low birth weight		
Preterm birth	49.67	[49.29, 50.05]
Panel D: Direct association of prenatal and post-birth PM_2.5_ exposure with infant death		
Prenatal exposure		
First trimester		
Ref: <8 μg/m^3^		
[8.00–10.00 μg/m^3^)	0.03	[0.01, 0.05]
[10.00–12.00 μg/m^3^)	0.04	[0.01, 0.06]
[12.00–19.16 μg/m^3^]	0.03	[0, 0.06]
Second trimester		
Ref: <8 μg/m^3^		
[8.00–10.00 μg/m^3^)	0.04	[0.02, 0.05]
[10.00–12.00 μg/m^3^)	0.04	[0.01, 0.06]
[12.00–19.16 μg/m^3^]	0.07	[0.04, 0.09]
Third trimester		
Ref: <8 μg/m^3^		
[8.00–10.00 μg/m^3^)	0.03	[0.02, 0.05]
[10.00–12.00 μg/m^3^)	0.07	[0.05, 0.09]
[12.00–19.16 μg/m^3^]	0.1	[0.06, 0.13]
Post-birth exposure		
Ref: <8 μg/m^3^		
[8.00–10.00 μg/m^3^)	0.04	[0.01, 0.07]
[10.00–12.00 μg/m^3^)	0	[−0.03, 0.04]
[12.00–17.19 μg/m^3^]	−0.01	[−0.07, 0.05]
Panel E: Direct association of preterm birth and low birth weight with infant death		
Preterm birth	2	[1.93, 2.07]
Low birth weight	3.64	[3.55, 3.73]
Number of observations	10,017,357	
Average SRMR	0	

Notes: All coefficients are expressed as percentage point changes in the respective outcomes. 95% confidence intervals were estimated using standard errors that were clustered at the county-level. The post-birth PM_2.5_ exposure is estimated over a 12-month period following the end of the nine-month prenatal period. SRMR = Standardized Root Mean Square Residual. The average SRMR was calculated as the average of the SRMR of the Structural Equation Model fit in each of the five imputed datasets.

**Table 3 ijerph-19-00258-t003:** Direct and indirect associations of prenatal and post-birth PM_2.5_ exposure (defined over 12 months) with infant mortality.

	Direct Association		Indirect Association		Total Association		Proportion Mediated (%)
	Percentage Point Change	95% Confidence Interval	Percentage Point Change	95% Confidence Interval	Percentage Point Change	95% Confidence Interval	
Panel A: Direct and indirect associations of the prenatal PM_2.5_ exposure							
First trimester							
Ref: <8 μg/m^3^							
[8.00–10.00 μg/m^3^)	0.03	[0.01, 0.05]	0.01	[0, 0.02]	0.04	[0.01, 0.06]	25%
[10.00–12.00 μg/m^3^)	0.04	[0.01, 0.06]	0.01	[0, 0.01]	0.04	[0.01, 0.07]	25%
[12.00–19.16 μg/m^3^]	0.03	[0, 0.06]	0	[−0.02, 0.01]	0.03	[−0.01, 0.07]	0%
Second trimester							
Ref: <8 μg/m^3^							
[8.00–10.00 μg/m^3^)	0.04	[0.02, 0.05]	0.02	[−0.01, 0.05]	0.05	[0.03, 0.07]	40%
[10.00–12.00 μg/m^3^)	0.04	[0.01, 0.06]	0.03	[−0.02, 0.08]	0.07	[0.04, 0.09]	43%
[12.00–19.16 μg/m^3^]	0.07	[0.04, 0.09]	0.04	[−0.03, 0.1]	0.1	[0.07, 0.13]	40%
Third trimester							
Ref: <8 μg/m^3^							
[8.00–10.00 μg/m^3^)	0.03	[0.02, 0.05]	0.02	[−0.01, 0.05]	0.05	[0.03, 0.07]	40%
[10.00–12.00 μg/m^3^)	0.07	[0.05, 0.09]	0.04	[−0.03, 0.1]	0.1	[0.08, 0.13]	40%
[12.00–19.16 μg/m^3^]	0.1	[0.06, 0.13]	0.05	[−0.04, 0.14]	0.15	[0.11, 0.18]	33%
Panel B: Direct and indirect associations of the postnatal PM_2.5_ exposure							
Ref: <8 μg/m^3^							
[8.00–10.00 μg/m^3^)	0.04	[0.01, 0.07]	-	-	0.04	[0.01, 0.07]	-
[10.00–12.00 μg/m^3^)	0	[−0.03, 0.04]	-	-	0	[−0.03, 0.04]	-
[12.00–19.16 μg/m^3^]	−0.01	[−0.07, 0.05]	-	-	−0.01	[−0.07, 0.05]	-

Notes: The indirect association from the prenatal PM_2.5_ exposure to infant mortality reflects two paths: prenatal exposure → preterm birth → infant mortality; and, prenatal exposure → preterm birth → low birth weight → infant mortality. The post-birth PM_2.5_ exposure is estimated over a 12-month period following the end of the nine-month prenatal period. Since there is only a direct path from the post-birth exposure to infant mortality, there are no results for the indirect association between the post-birth exposure and the outcome.

## Data Availability

We used restricted-access, cohort-linked live birth–infant death data provided by the National Center for Health Statistics (NCHS). Readers who would like to use these data must apply to the NCHS separately. All other datasets used in this analysis are publicly available from various websites. We have noted the websites from which the data were downloaded in references [21,24,25,26,27,28].

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
