# Peer review of "Understanding the Pathways from Prenatal and Post-Birth PM2.5 Exposure to Infant Death: An Observational Analysis Using US Vital Records (2011–2013)"

_ijerph, 2021, doi:10.3390/ijerph19010258_

Round 1
Reviewer 1 Report
line 31 "in mixed" should be "is mixed"?!;
Definitions in lines 99 till 146 could be presented in brief topics, perhaps in a table);
"Appendix Figure A1 Graphical representation of the Structural Equation Model."....should appear before (in the paper). to help readers.
General Comment/question: Despite the "big data"/number of infants, how representative is the medium time interval average (trimester/annual) PM2.5 concentrations on the development of the babies? According to figure 1, PM2.5, even considering the 4 years, only varies between 3.9 and 15.46 ug/m3 (average 12 months). How significant is this range for Health issues? Another question is if indoor air quality can be significantly different form the ambient air, and if this difference can be more strong in south regions due to more "house ventilation" due to better weather.
Overall, the work presents conclusions based on statistical analysis from observational data.
Reviewer 2 Report
This interesting paper examined the associations of pre- and post-natal exposure to PM2.5 with the risk of infant death for all births which took place in the US between 2011 and 2013 and found a direct positive association between air pollution and infant death.
The authors report that their approach to the issue takes a different line (structural equation modelling) from other studies which enables them to examine the effect of possible confounders more clearly.
Comments:
"dismantle" or "disassemble" are more common usages rather than "decompose" in the kind of examination the authors undertook.
Line 31 - "in" = "is".
Line 87-88, surely a "pregnant woman"?
Line 129, what is the "birth certificate revision" of 2003?
Line 162, "plural birth"; I am more used to the term "multiple birth"
Line 175, what do you mean by "demeaning the data at county level"?
Discussion, page 4, 2nd paragraph: what do you mean by "net out"?
Discussion, page 4: movement of pregnant women - what kind of movement do you mean? E.g., moving residence, or travelling on visits, or what?
Since the authors use "prenatal" I am surprised at "post-birth" when "postnatal" is in common usage.
One decimal point is sufficient throughout the MS; with a long table keeping percentages to whole numbers can enhance readability; anyone wanting data to one decimal point can calculate that themselves.
The authors may be interested in the historical review of the London Smog of the early 1950s which shows a parallel picture to their findings: Bharadwaj et al. 2016. ‘Early-Life Exposure to the Great Smog of 1952 and the Development of Asthma’. American Journal of Respiratory and Critical Care Medicine 194 (12): 1475–82. https://doi.org/10.1164/rccm.201603-0451OC.
